# ExPT: Synthetic Pretraining for Few-Shot Experimental Design

**Tung Nguyen, Sudhanshu Agrawal, Aditya Grover**
University of California, Los Angeles
{tungnd,adityag}@cs.ucla.edu, sudhanshuagr27@g.ucla.edu

## Abstract

Experimental design for optimizing black-box functions is a fundamental problem in many science and engineering fields. In this problem, sample efficiency is crucial due to the time, money, and safety costs of real-world design evaluations. Existing approaches either rely on active data collection or access to large, labeled datasets of past experiments, making them impractical in many real-world scenarios. In this work, we address the more challenging yet realistic setting of *few-shot* experimental design, where only a few labeled data points of input designs and their corresponding values are available. We introduce **Ex**periment **P**retrained **T**ransformers (ExPT), a foundation model for few-shot experimental design that combines unsupervised learning and in-context pretraining. In ExPT, we only assume knowledge of a finite collection of unlabelled data points from the input domain and pretrain a transformer neural network to optimize diverse synthetic functions defined over this domain. Unsupervised pretraining allows ExPT to adapt to any design task at test time in an in-context fashion by conditioning on a few labeled data points from the target task and generating the candidate optima. We evaluate ExPT on few-shot experimental design in challenging domains and demonstrate its superior generality and performance compared to existing methods. The source code is available at `https://github.com/tung-nd/ExPT.git`.

## 1 Introduction

The design of experiments to optimize downstream target objectives is a ubiquitous challenge across many science and engineering domains, including materials discovery [27], protein engineering [7, 49, 2], molecular [22] design, mechanical design [4, 38], and neural architecture optimization [66]. The key criterion of interest in experimental design (ED) is sample-efficiency, as the target objectives are often black-box functions and evaluating these objectives for any candidate design often involves expensive real-world experiments. A standard class of approaches learn a surrogate to approximate the target objective and actively improve the approximation quality through online experiments [53]. However, online data acquisition may be infeasible in the real world due to high costs, time constraints, or safety concerns. As an alternate, recent works have proposed offline ED [7, 58, 36, 57, 12, 34], wherein a model learns to perform optimization from a fixed dataset of past experiments. While this is more practical than the online setting, current offline methods and benchmarks assume access to large experimental datasets containing thousands of data points, which are hard or even impossible to obtain in high-stake and emerging science problems. Even when these datasets exist, the past experiments might be of very poor quality resulting in poor surrogate learning and optimization.

In this paper, we aim to overcome these limitations for hyper-efficient experimental design that does not require large experimental datasets. To this end, we introduce *few-shot* experimental design, a more challenging setting that better resembles real-world scenarios. We describe few-shot ED as a two-phased pretraining-adaptation paradigm. In the pretraining phase, we only assume access to

37th Conference on Neural Information Processing Systems (NeurIPS 2023).

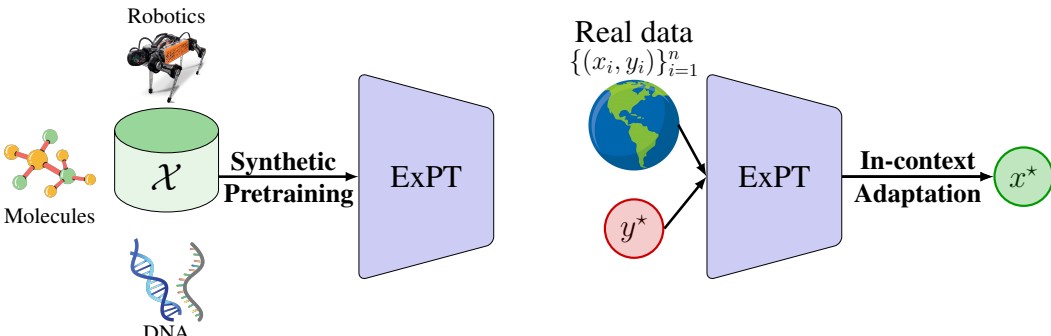

Figure 1: Experiment Pretrained Transformers (ExPT) follow a pretraining-adaptation approach for few-shot experimental design. During *pretraining* (**left**), the model has access to unlabeled designs from domain $\mathcal{X}$ without their corresponding scores. For *adaptation* (**right**), the model conditions on a small set of (design, score) pairs and the desired score $y^\star$ to generate the optimal design $x^\star$.

*unlabeled* data, i.e., input designs without associated function values. During the adaptation phase, we have access to a few labeled examples of past experiments to adapt the model to the downstream task. This setup offers several advantages. First, it alleviates the requirement for costly annotated data and relies mainly on unlabeled inputs that are easily accessible. Second, unsupervised pretraining enables us to utilize the same pretrained backbone for adapting to multiple downstream optimization tasks within the same domain. For example, in molecule design, one may want to optimize for multiple properties, including drug-likeness, synthesizability, or similarity to target molecules [8, 17].

The key question in this setup is how to make use of the unlabeled data to facilitate efficient generalization to downstream tasks during optimization. Our intuition here is that, while the objective function is unknown, we can use the unlabeled inputs to generate pretraining data from other *synthetic* functions. If a model can few-shot learn from a diverse and challenging set of functions, it should be able to generalize quickly to any target objective during the adaptation phase, in line with recent foundation models for language [9] and vision [3]. This insight gives rise to our idea of *synthetic pretraining*, wherein we pretrain the model on data generated from a rich family of synthetic functions that operate on the same domain as the target task. Specifically, for each function drawn from this family, we sample a set of points by using the unlabeled data as inputs. We divide these points into a small context set and a target set, and train the model via in-context learning to perform conditional generation of the target input $x$ given the context points and the target value $y$. A model that works well on this task should be able to efficiently capture the structures of the underlying function, i.e., how different regions of the input space influence the function value, from a small context set. By explicitly training the model to perform this task on a diverse set of functions, the model can generalize efficiently to downstream functions during adaptation requiring only limited supervision. After pretraining, we can perform optimization by conditioning the model on a few labeled examples from the downstream task and generating an input that achieves the optimum $y^\star$.

Inspired by recent advances in few-shot learning in language [9, 42] and other domains [44, 45, 19, 37], we instantiate a novel foundation model with a transformer-based architecture [59], which we call **Ex**periment **P**retrained **T**ransformers (ExPT). ExPT is an encoder-decoder architecture, in which the encoder is a transformer [59] network that encodes the context points and the target value, and the decoder is a VAE [32] model that predicts the high-dimensional target input. The transformer encoder allows ExPT to perform few-shot generation and optimization purely through in-context learning in a gradient-free fashion. We compare the performance of ExPT and various baselines on 2 few-shot settings created from Design-Bench [58], a standard database benchmark for ED. The two settings allow us to examine how different methods perform with respect to different quantities and qualities of few-shot data. In both these settings, results show that ExPT achieves the highest average score and the highest average ranking with respect to median performance, mean performance, and best-achieved performance. Especially in the more challenging setting, ExPT outperforms the second-best method by 70% in terms of the mean performance. Additionally, we explore the potential of using the same pretrained ExPT for multiple objectives, and conduct extensive ablation studies to validate the effectiveness of our design choices for synthetic data generation and ExPT architecture.

## 2 Experiment Pretrained Transformers

### 2.1 Problem setup

Let $f : \mathcal{X} \to \mathbb{R}$ be a black-box function that operates on a $d$-dimensional domain $\mathcal{X} \subseteq \mathbb{R}^d$. In experimental design (ED), the goal is to find the input $x^\star$ that maximizes $f$:

$$x^\star \in \arg\max_{x \in \mathcal{X}} f(x). \tag{1}$$

Typically, $f$ is a high-dimensional and complex function that often involves expensive physical experiments. Existing approaches either assume the ability to actively query $f$ to collect data [53] or access to a large dataset of past experiments [58]. Both assumptions are too strong in many real-world applications where data collection is hard or even impossible [11]. Therefore, we propose *few-shot ED*, a more challenging yet realistic setting to overcome these limitations. In few-shot ED, the goal is to optimize *any* objective function in the domain $\mathcal{X}$ given only a handful of examples. We approach this problem with a pretraining-adaptation pipeline. In the *pretraining* phase, we assume access to an *unlabeled* dataset $\mathcal{D}_{\text{unlabeled}} = \{x_i\}_{i=1}^{|\mathcal{D}_{\text{unlabeled}}|}$ from the optimization domain $\mathcal{X} \subseteq \mathbb{R}^d$. We note that $\mathcal{D}_{\text{unlabeled}}$ only contains potential design inputs without their corresponding scores, for example, potential molecules in molecule optimization or different combinations of hyperparameters in neural architecture search. This means the objective function $f$ is *unspecified* during pretraining.

During the *adaptation* phase, one can use the pretrained model to optimize any objective function $f$ in the same domain $\mathcal{X}$. We now have access to a few-shot *labeled* dataset that the model can use to adapt to the downstream function $\mathcal{D}_{\text{few-shot}} = \{(x_1, y_1), \ldots, (x_n, y_n)\}$, in which $y_i = f(x_i)$ and $n = |\mathcal{D}_{\text{few-shot}}| \ll |\mathcal{D}_{\text{unlabeled}}|$. After adaptation, we evaluate a few-shot optimization method by allowing it to propose $Q$ input $x's$ and query their scores using the black-box function $f$, where $Q$ is often called the optimization budget [58, 36, 57, 12, 34]. The performance of a black-box optimizer is then measured by computing the max, median, and mean of the $Q$ evaluations, This setup provides two key benefits. First, it resembles many real-world scenarios, where the potential design inputs are cheap and easy to obtain while their target function values are expensive to evaluate. For example, in molecular optimization, we have databases of millions of molecules [30, 5, 48] but only the properties of a handful are known [28, 47, 23]. Second, unsupervised pretraining allows us to train a general backbone that we can adapt to multiple optimization tasks in the same domain.

### 2.2 Synthetic Pretraining and Inverse Modeling for Scalable Experimental Design

Intuitively, the adaptation phase in §2.1 resembles a few-shot learning problem, in which a model is tasked to produce the optimal input $x^\star$ by conditioning on a few labeled examples in $\mathcal{D}_{\text{few-shot}}$. To perform well in this task, a model has to efficiently capture the structure of a high-dimension function $f$, i.e., what regions of the function lead to higher values and vice versa, from very few examples in $\mathcal{D}_{\text{few-shot}}$. Given this perspective, the question now is how to make use of the unlabeled dataset $\mathcal{D}_{\text{unlabeled}}$ to pretrain a model that achieves such efficient generalization to the objective function $f$. Our key insight is, if a model learns to perform in-context learning on a diverse and challenging set of functions, it should be able to adapt quickly to any objective function at test time. While the function values are unknown during pretraining, we can use the unlabeled inputs $x's$ to generate pretraining data from *other* functions. This gives rise to our idea of *synthetic pretraining*, wherein we pretrain the model to perform few-shot learning on a family of synthetic functions $\tilde{F}$ that operate on the same input domain $\mathcal{X}$ of the objective $f$. We discuss in detail our mechanism for synthetic data generation in Section 2.3. For each function $\tilde{f}$ generated from $\tilde{F}$, we sample a set of function evaluations $\{(x_i, y_i)\}_{i=1}^N$ that we divide into a small context set $\{(x_i, y_i)\}_{i=1}^m$ and a target set $\{(x_j, y_j)\}_{j=m+1}^N$. We train the model to predict the target points conditioning on the context set.

There are two different approaches to pretraining a model on this synthetic data. The first possible approach is forward modeling, where the model is trained to predict the target outputs $y_{m+1:N}$ given the context points and the target inputs $x_{m+1:N}$. This is similar to the approach followed by TNPs [45], a model recently proposed in the context of meta-learning. During adaptation, we can condition the model on the labeled examples in $\mathcal{D}_{\text{few-shot}}$ and perform gradient ascent updates to improve an existing design input $x_t$. However, as commonly observed in previous works [58, 36, 57], this approach is susceptible to producing highly suboptimal inputs. This is because performing gradient ascent with respect to an imperfect forward model may result in points that have high values

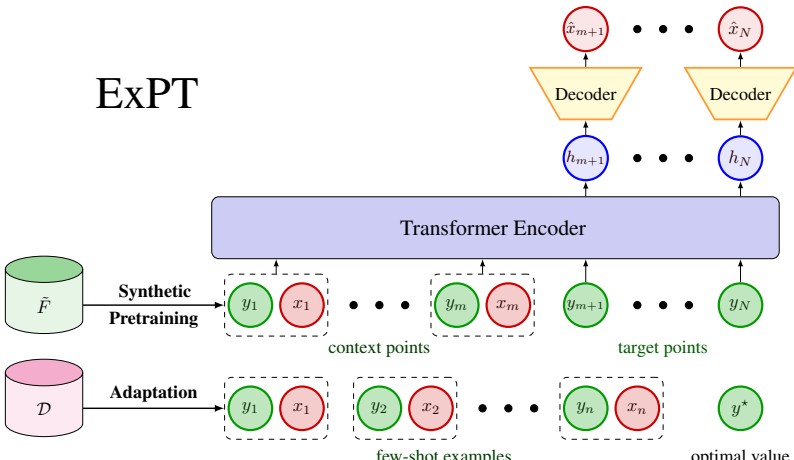

Figure 2: The pretraining-adaptation phases for ExPT. We sample synthetic data from $\tilde{F}$ and pretrain the model to maximize $\log p(x_{m+1:N} \mid x_{1:m}, y_{1:m}, y_{m+1:N})$. At adaptation, the model conditions on $\mathcal{D}_{\text{few-shot}}$ and $y^\star$ to generate candidates. ExPT employs a transformer encoder that encodes the context points and target outputs and a relevant decoder that predicts the target inputs.

under the model but are poor when evaluated using the real function. Instead, we propose to perform *inverse modeling*, where the model learns to predict the inputs $x_{m+1:N}$ given the output values $y_{m+1:N}$ and the context points. As the model learns to directly generate input $x's$, it is less vulnerable to the aforementioned problem. Another advantage of inverse modeling is after pretraining, we can simply condition on $\mathcal{D}_{\text{few-shot}}$ and the optimal value $y^\star$ to generate the candidate optima. Our loss function for pretraining the model is:

$$
\theta = \arg\max_{\theta} \mathbb{E}_{\tilde{f}\sim\tilde{F}, x_{1:N}\sim\mathcal{D}_{\text{unlabeled}}, y_{1:N}=\tilde{f}(x_{1:N})} \left[\log p(x_{m+1:N} \mid x_{1:m}, y_{1:m}, y_{m+1:N})\right]
$$

$$
= \arg\max_{\theta} \mathbb{E}_{\tilde{f}\sim\tilde{F}, x_{1:N}\sim\mathcal{D}_{\text{unlabeled}}, y_{1:N}=\tilde{f}(x_{1:N})} \left[\sum_{i=m+1}^{N} \log p(x_i \mid x_{1:m}, y_{1:m}, y_i)\right], \tag{2}
$$

where we assume the target points are independent given the context set and the target output. Figure 2 illustrates the proposed pretraining and adaptation pipeline. Typically, we use a small context size $m$ during pretraining to resemble the test scenario.

After pretraining, ExPT can adapt to any objective $f$ in the domain in a gradient-free fashion. Samples in the few-shot dataset $\mathcal{D}_{\text{few-shot}}$ become the context points and the model conditions on only one target $y^\star$, which is the optimal value of $f$, to generate candidate optima. Note that we only assume the knowledge of $y^\star$ and not $x^\star$. This assumption is common in many prior works [34, 46, 13, 14]. In practice, $y^\star$ might be known based on domain knowledge. For example, in molecule design, there are physical limits on the value of certain properties such as relaxed energy, in robot applications, the optimal performance can be computed from the cost function, and in neural architecture search, we know the theoretical limits on the highest possible accuracy for classifiers.

Next, we present the details of synthetic data generation and our proposed model architecture, the two components that constitute our proposed foundation model, which we refer to as **Ex**periment **P**retrained **T**ransformers (ExPT).

## 2.3   Data generation

We need a family of functions to generate synthetic data for pretraining ExPT. A good family of functions should be easy to sample from and should be capable of producing diverse functions. Many possible candidates exist for synthetic function families, such as Gaussian Processes (GPs), randomly constructed Gaussian Mixture Models, or randomly initialized or pretrained neural networks. Among these candidates, we choose to generate synthetic data from Gaussian Processes with an RBF kernel. This is for several reasons. First, they are a natural choice as they represent distributions over functions. Second, it is easy and cheap to sample data from prior GPs. And third, a GP with an RBF

kernel is a universal approximator to any function [41]. Specifically, $\tilde{f}$ is sampled as follows,

$$\tilde{f} \sim \mathcal{GP}(0, \mathcal{K}), \quad \mathcal{K}(x, x') = \sigma^2 \exp\left(-\frac{(x-x')^2}{2\ell^2}\right), \tag{3}$$

in which $\sigma$ and $\ell$ are the two hyperparameters of the RBF kernel. The variance $\sigma$ scales the magnitudes of the covariance matrix. A larger variance results in a wider range of function values, while a smaller variance restricts the function values to a narrower range. On the other hand, the length scale $\ell$ determines how strongly the covariance matrix varies with respect to the distance between $x$ and $x'$. A very small length scale $\ell$ means the kernel is sensitive to the distance between $x$ and $x'$, leading to sharp transitions between neighboring points and a lack of smoothness in the functions. In contrast, if $\ell$ is too large, the covariance between points will be similar for both near and distant points, leading to function values that are very similar. In other words, too large a length scale reduces the diversity of the synthetic functions. In practice, we randomize both $\sigma$ and $\ell$ to increase the diversity of the pretraining data. Appendix C.1 demonstrates the empirical importance of these hyperparameters.

## 2.4 Model architecture

To optimize the loss function in Equation (2), we need a model architecture that can condition on a few examples drawn from an underlying function to make predictions for other points. This resembles the idea of *in-context learning* that has proven very successful in language [9, 42] and other domains [44, 45, 19, 37]. The key to success in these works is a transformer architecture that performs in-context learning efficiently via the attention mechanism [59]. Inspired by this, we instantiate ExPT with a transformer-based architecture. Figure 2 illustrates the ExPT overall architecture. Specifically, ExPT employs a transformer encoder that encodes the context points $\{(x_i, y_i)\}_{i=1}^m$ and the target inputs $y_{m+1:N}$, and outputs hidden vectors $h_{m+1:N}$. To inform the model that $x_i$ and $y_i$ are the input and the corresponding output from $\tilde{f}$, we concatenate them to form a token. This results in the sequence $\{(y_1, x_1), \ldots, (y_m, x_m), y_{m+1}, \ldots, y_N\}$. We then embed these tokens using two 1-layer MLP networks, one for the pairs and one for the target inputs, before feeding the sequence to the transformer layers. We implement a masking mechanism that prevents the context points from attending the target points, as they do not contain information about the underlying function $\tilde{f}$.

Each hidden vector $h_i$ output by the transformer encoder encompasses the information of the context points and the target input $y_i$. Therefore, given $h_i$, the conditional probability $p_\theta(x_i \mid x_{1:m}, y_{1:m}, y_i)$ reduces to $p_\theta(x_i \mid h_i)$. As $x_i$ is high-dimensional, we can utilize existing generative models to model the conditional distribution $p(x_i \mid h_i)$. In this work, we train a conditional VAE model [32] alongside the transformer encoder because of its training stability, light hyperparameter tuning, and good empirical performance. For discrete tasks, we follow the same procedure as Trabucco et al. [58] that emulates logit values by interpolating between a uniform distribution and the one hot values. We train the entire model by maximizing the lower bound of the conditional likelihood $\log p(x_i \mid h_i)$:

$$\log p_\theta(x_i \mid h_i) \geq \mathbb{E}_{q_\phi(z \mid x_i, h_i)}\left[\log p_\theta(x_i \mid z, h_i)\right] - \mathrm{KL}(q_\phi(z \mid x_i, h_i) \| p(z)), \tag{4}$$

in which $q_\phi(z \mid x_i, h_i)$ is the encoder of the conditional VAE and $p(z)$ is a standard Gaussian prior.

## 3 Experiments

### 3.1 Synthetic experiments

We first evaluate the performance of ExPT in a synthetic experiment, where we train ExPT on data generated from Gaussian Processes (GPs) with an RBF kernel and test the model on four out-of-distribution functions drawn from four different kernels: Matern, Linear, Cosine, and Periodic. Figure 3 shows the performance of ExPT on four test functions through the course of training. The model performs well on all four functions, achieving scores that are much higher than the max value in the few-shot dataset, and approaching the true optimal value, even for kernels that are significantly different from RBF like Cosine and Periodic. Moreover, the performance improves consistently as we pretrain, showing that pretraining facilitates generalization to out-of-distribution functions with very few examples. See Appendix A for a detailed setup of this experiment.

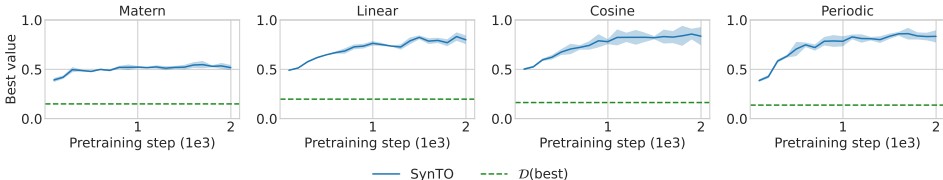

Figure 3: The performance of ExPT on $4$ out-of-distribution synthetic tasks through the pretraining phase. We average the performance across $3$ seeds.

## 3.2 Design-Bench experiments

**Tasks** We consider $4$ tasks from Design-Bench[1] [58]. **D'Kitty** and **Ant** are continuous tasks with input dimensions of $56$ and $60$, respectively. In D'kitty and Ant, the goal is to optimize the morphological structure of two simulated robots, Ant [6] to run as fast as possible, and D'kitty [1] to reach a fixed target location. **TF Bind 8** and **TF Bind 10** are two discrete tasks, where the goal is to find the length-$8$ and length-$10$ DNA sequence that has a maximum binding affinity with the SIX6_REF_R1 transcription factor. The design space in these two tasks consists of sequences of one of four categorical variables, corresponding to four types of nucleotide. For each task, Design-Bench provides a `public` dataset, a larger `hidden` dataset which is used to normalize the scores, and an oracle. We have an exact oracle to evaluate the proposed designs in all $4$ tasks we consider.

**Few-shot settings** We create 2 few-shot settings from the above tasks, which we call `random` and `poorest`. In `random`, we randomly subsample $1\%$ of data points in the `public` set of each task as the few-shot dataset $\mathcal{D}_{\text{few-shot}}$. The `poorest` setting is more challenging, where we use $1\%$ of the data points which have the *lowest* scores. The two settings examine how sensitive different methods are to the quantity and quality of the data. In both settings, we use $x's$ in the `public` dataset as $\mathcal{D}_{\text{unlabeled}}$.

**ExPT details** For each domain, we pretrain ExPT for $10,000$ iterations with $128$ synthetic functions in each iteration, corresponding to a total number of $1,280,000$ synthetic functions. For each function, we randomly sample $228$ input $x's$ from the unlabeled dataset $\mathcal{D}_{\text{unlabeled}}$ and generate the values $y's$ from a Gaussian Process with an RBF kernel. To increase the diversity of synthetic data, we randomize the two hyperparameters, length scale $\ell \sim \mathcal{U}[5.0, 10.0]$ and function scale $\sigma \sim \mathcal{U}[1.0, 1.0]$, when generating each function. Additionally, we add Gaussian noises $\epsilon \sim \mathcal{N}(0, 0.1)$ to each input $x$ sampled from $\mathcal{D}_{\text{unlabeled}}$ to enlarge the pretraining inputs. For each generated function, we use $100$ points as context points and the remaining $128$ as target points, and train the model to optimize (2). During the adaptation phase, we condition the pretrained ExPT model on the labeled few-shot dataset $\mathcal{D}_{\text{few-shot}}$ and the target function value $y^\star$ to generate designs $x's$.

**Baselines** We compare ExPT with BayesOpt (GP-qEI) [53], a canonical ED method, and MINs [36], COMs [57], BDI[12], and BONET [34], four recent deep learning models that have achieved state-of-the-art performance in the offline setting. To adapt GP-qEI to the few-shot setting, we use a feedforward network trained on few-shot data to serve as an oracle, a Gaussian Process to quantify uncertainty, and the quasi-Expected Improvement [61] algorithm for the acquisition function. For the deep learning baselines, we train their models on the few-shot dataset $\mathcal{D}_{\text{few-shot}}$ using the hyperparameters reported in their original papers.

**Evaluation** For each considered method, we allow an optimization budget $Q = 256$. We report the median score, the max score, and the mean score among the $256$ proposed inputs. Following previous works, we normalize the score to $[0, 1]$ by using the minimum and maximum function values from a large hidden dataset $y_{\text{norm}} = \frac{y - y_{\text{min}}}{y_{\text{max}} - y_{\text{min}}}$. We report the mean and standard deviation of the score across $3$ independent runs for each method.

**Results** Table 1 shows the performance of different methods in the `random` setting. Most methods perform well in the `random` setting, where ExPT achieves the highest average score and the best average rank across all $3$ performance metrics. For each of the tasks and metrics considered, ExPT is either the best or second-best performing method. Notably, in Ant, ExPT significantly outperforms the best baseline by $18\%$, $9\%$, and $10\%$ with respect to the median, max, and mean performance,

---

[1]We exclude domains where the oracle functions are flagged to be highly inaccurate and noisy in prior works (ChEMBL, Hopper, and Superconductor), or too expensive to evaluate (NAS). See Appendix B for more details.

Table 1: Comparison of ExPT and the baselines on the few-shot `random` setting of 5 Design-Bench tasks. We report median, max, and mean performance across 3 random seeds. Higher scores and lower ranks are better. Blue denotes the best entry in the column, and Violet denotes the second best.

| | Baseline | D'Kitty | Ant | TF Bind 8 | TF Bind 10 | Mean score (↑) | Mean rank (↓) |
|---|---|---|---|---|---|---|---|
| | $\mathcal{D}_{\text{few-shot}}$(best) | 0.883 | 0.563 | 0.439 | 0.466 | — | — |
| Median | MINs | $0.859 \pm 0.014$ | $0.485 \pm 0.152$ | $0.416 \pm 0.019$ | $0.468 \pm 0.014$ | $0.557 \pm 0.050$ | 4.0 |
| | COMs | $0.752 \pm 0.007$ | $0.411 \pm 0.012$ | $0.371 \pm 0.001$ | $0.468 \pm 0.000$ | $0.501 \pm 0.005$ | 4.0 |
| | BONET | $0.852 \pm 0.013$ | $0.597 \pm 0.119$ | $0.441 \pm 0.003$ | $0.483 \pm 0.009$ | $0.593 \pm 0.036$ | 2.3 |
| | BDI | $0.592 \pm 0.020$ | $0.396 \pm 0.018$ | $0.540 \pm 0.032$ | $0.438 \pm 0.034$ | $0.492 \pm 0.026$ | 4.8 |
| | GP-qEI | $0.842 \pm 0.058$ | $0.550 \pm 0.007$ | $0.439 \pm 0.000$ | $0.467 \pm 0.000$ | $0.575 \pm 0.016$ | 4.0 |
| | ExPT | $0.902 \pm 0.006$ | $0.705 \pm 0.018$ | $0.473 \pm 0.014$ | $0.477 \pm 0.014$ | $0.639 \pm 0.013$ | 1.5 |
| Max | MINs | $0.930 \pm 0.010$ | $0.890 \pm 0.017$ | $0.814 \pm 0.030$ | $0.639 \pm 0.017$ | $0.818 \pm 0.019$ | 3.3 |
| | COMs | $0.920 \pm 0.010$ | $0.841 \pm 0.044$ | $0.686 \pm 0.152$ | $0.656 \pm 0.020$ | $0.776 \pm 0.057$ | 4.0 |
| | BONET | $0.909 \pm 0.012$ | $0.888 \pm 0.024$ | $0.887 \pm 0.053$ | $0.702 \pm 0.006$ | $0.847 \pm 0.024$ | 3.0 |
| | BDI | $0.918 \pm 0.006$ | $0.806 \pm 0.094$ | $0.906 \pm 0.074$ | $0.532 \pm 0.023$ | $0.791 \pm 0.049$ | 4.5 |
| | GP-qEI | $0.896 \pm 0.000$ | $0.887 \pm 0.000$ | $0.513 \pm 0.104$ | $0.647 \pm 0.011$ | $0.736 \pm 0.029$ | 5.0 |
| | ExPT | $0.973 \pm 0.005$ | $0.970 \pm 0.048$ | $0.933 \pm 0.036$ | $0.677 \pm 0.048$ | $0.888 \pm 0.023$ | 1.3 |
| Mean | MINs | $0.624 \pm 0.025$ | $0.009 \pm 0.013$ | $0.415 \pm 0.030$ | $0.465 \pm 0.015$ | $0.378 \pm 0.021$ | 4.8 |
| | COMs | $0.515 \pm 0.050$ | $0.020 \pm 0.006$ | $0.369 \pm 0.003$ | $0.471 \pm 0.004$ | $0.344 \pm 0.016$ | 4.8 |
| | BONET | $0.837 \pm 0.023$ | $0.579 \pm 0.024$ | $0.448 \pm 0.011$ | $0.484 \pm 0.009$ | $0.587 \pm 0.017$ | 2.0 |
| | BDI | $0.570 \pm 0.032$ | $0.385 \pm 0.012$ | $0.536 \pm 0.032$ | $0.444 \pm 0.027$ | $0.484 \pm 0.026$ | 3.5 |
| | GP-qEI | $0.505 \pm 0.006$ | $0.019 \pm 0.001$ | $0.439 \pm 0.001$ | $0.473 \pm 0.002$ | $0.359 \pm 0.003$ | 4.5 |
| | ExPT | $0.865 \pm 0.016$ | $0.639 \pm 0.026$ | $0.476 \pm 0.010$ | $0.474 \pm 0.015$ | $0.614 \pm 0.017$ | 1.5 |

Table 2: Comparison of ExPT and the baselines on the few-shot `poorest` setting of 5 Design-Bench tasks. We report the median, max, and mean performance across 3 random seeds. Higher scores and lower ranks are better. Blue denotes the best entry in the column, and Violet denotes the second best.

| | Baseline | D'Kitty | Ant | TF Bind 8 | TF Bind 10 | Mean score (↑) | Mean rank (↓) |
|---|---|---|---|---|---|---|---|
| | $\mathcal{D}_{\text{few-shot}}$(best) | 0.307 | 0.124 | 0.124 | 0.000 | — | — |
| Median | MINs | $0.480 \pm 0.156$ | $0.316 \pm 0.040$ | $0.437 \pm 0.007$ | $0.463 \pm 0.003$ | $0.424 \pm 0.052$ | 3.5 |
| | COMs | $0.733 \pm 0.023$ | $0.401 \pm 0.026$ | $0.111 \pm 0.000$ | $0.459 \pm 0.006$ | $0.426 \pm 0.014$ | 4.3 |
| | BONET | $0.310 \pm 0.000$ | $0.236 \pm 0.047$ | $0.319 \pm 0.018$ | $0.461 \pm 0.017^*$ | $0.332 \pm 0.021$ | 4.8 |
| | BDI | $0.309 \pm 0.000$ | $0.192 \pm 0.012$ | $0.365 \pm 0.000$ | $0.454 \pm 0.017$ | $0.330 \pm 0.007$ | 5.5 |
| | GP-qEI | $0.883 \pm 0.000$ | $0.565 \pm 0.001$ | $0.439 \pm 0.000$ | $0.467 \pm 0.000$ | $0.589 \pm 0.000$ | 2.0 |
| | ExPT | $0.922 \pm 0.009$ | $0.686 \pm 0.090$ | $0.552 \pm 0.042$ | $0.489 \pm 0.013$ | $0.662 \pm 0.039$ | 1.0 |
| Max | MINs | $0.841 \pm 0.014$ | $0.721 \pm 0.031$ | $0.962 \pm 0.025$ | $0.648 \pm 0.025$ | $0.793 \pm 0.022$ | 3.3 |
| | COMs | $0.931 \pm 0.022$ | $0.843 \pm 0.020$ | $0.124 \pm 0.000$ | $0.739 \pm 0.057$ | $0.659 \pm 0.025$ | 3.3 |
| | BONET | $0.929 \pm 0.031$ | $0.557 \pm 0.118$ | $0.809 \pm 0.038$ | $0.519 \pm 0.039^*$ | $0.704 \pm 0.057$ | 5.0 |
| | BDI | $0.939 \pm 0.002$ | $0.693 \pm 0.109$ | $0.913 \pm 0.000$ | $0.596 \pm 0.020$ | $0.785 \pm 0.033$ | 3.5 |
| | GP-qEI | $0.896 \pm 0.000$ | $0.887 \pm 0.000$ | $0.439 \pm 0.000$ | $0.645 \pm 0.021$ | $0.717 \pm 0.005$ | 3.8 |
| | ExPT | $0.946 \pm 0.018$ | $0.965 \pm 0.004$ | $0.873 \pm 0.035$ | $0.615 \pm 0.022$ | $0.850 \pm 0.020$ | 2.3 |
| Mean | MINs | $0.623 \pm 0.051$ | $0.015 \pm 0.017$ | $0.464 \pm 0.009$ | $0.463 \pm 0.002$ | $0.391 \pm 0.020$ | 3.3 |
| | COMs | $0.607 \pm 0.021$ | $0.033 \pm 0.003$ | $0.109 \pm 0.001$ | $0.454 \pm 0.004$ | $0.301 \pm 0.007$ | 4.8 |
| | BONET | $0.490 \pm 0.023$ | $0.234 \pm 0.052$ | $0.318 \pm 0.018$ | $0.459 \pm 0.018$ | $0.375 \pm 0.028$ | 4.0 |
| | BDI | $0.364 \pm 0.004$ | $0.215 \pm 0.021$ | $0.369 \pm 0.000$ | $0.453 \pm 0.018$ | $0.350 \pm 0.011$ | 4.8 |
| | GP-qEI | $0.533 \pm 0.001$ | $0.018 \pm 0.000$ | $0.439 \pm 0.000$ | $0.470 \pm 0.002$ | $0.365 \pm 0.001$ | 3.5 |
| | ExPT | $0.871 \pm 0.018$ | $0.646 \pm 0.061$ | $0.549 \pm 0.032$ | $0.488 \pm 0.011$ | $0.639 \pm 0.031$ | 1.0 |

respectively. Only ExPT and BONET achieve a meaningful performance in Ant when considering the mean score. BONET is also the overall second-best method in this setting.

Table 2 shows the superior performance of ExPT in few-shot `poorest`, the more challenging setting. ExPT achieves the highest score in $10/12$ individual tasks and metrics, and also achieves the highest score and the best rank across tasks on average. Notably, in terms of the mean score, ExPT beats the best baseline by a large margin, achieving an improvement of $40\%$, $176\%$, and $18\%$ on D'Kitty, Ant, and TF Bind 8, and $70\%$ on average. The performance of most baselines drops significantly from the `random` to the `poorest` setting, including BONET, the second-best method in the `random` setting. This was also previously observed in the BONET paper [34]. Interestingly, the performance of ExPT, MINs, and GP-qEI is not affected much by the quality of the few-shot data, and even improves in certain metrics. We hypothesize that even though the dataset is of lower quality, it may contain specific anti-correlation patterns about the problem that the model can exploit.

**Pretraining analysis** In addition to the absolute performance, we investigate the performance of ExPT on downstream tasks through the course of pretraining. Figure 4 shows that the performance of ExPT in most tasks improves consistently as the number of pretraining steps increases. This shows that synthetic pretraining on diverse functions facilitates the generalization to complex real-world functions. In Ant, the performance slightly drops between 4000 and 10000 iterations. Therefore, we can further improve ExPT if we have a way to stop pretraining at a point that likely leads to the best

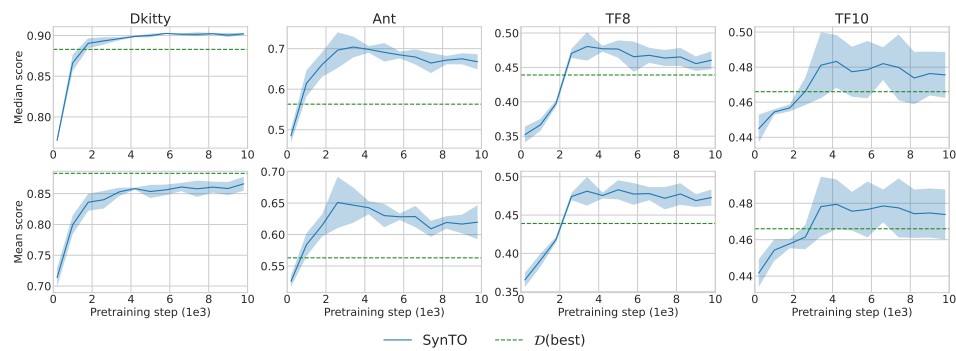

Figure 4: The median and mean performance of ExPT of 4 Design-Bench tasks through the course of pretraining. We average the performance across 3 seeds.

Table 3: ExPT's performance on different objectives in D'Kitty and Ant domains. We pretrain one model for all tasks in the same domain. The performance is averaged across 3 seeds.

|  | Task | D'Kitty | D'Kitty-45 | D'Kitty-60 | Ant | Ant-$v_y$ | Ant-Energy |
|---|---|---|---|---|---|---|---|
|  | $\mathcal{D}_{\text{few-shot}}$(best) | 0.307 | 0.297 | 0.344 | 0.124 | 0.210 | 0.189 |
| Median | ExPT | $0.922 \pm 0.009$ | $0.611 \pm 0.007$ | $0.569 \pm 0.010$ | $0.686 \pm 0.090$ | $0.613 \pm 0.009$ | $0.635 \pm 0.028$ |
| Max | ExPT | $0.976 \pm 0.004$ | $0.954 \pm 0.008$ | $0.973 \pm 0.004$ | $0.965 \pm 0.004$ | $0.923 \pm 0.049$ | $0.950 \pm 0.033$ |
| Mean | ExPT | $0.871 \pm 0.018$ | $0.619 \pm 0.016$ | $0.584 \pm 0.008$ | $0.646 \pm 0.061$ | $0.599 \pm 0.005$ | $0.608 \pm 0.025$ |

performance in downstream tasks. However, in practice, we do not have the luxury of testing on real functions during pretraining. Alternatively, we could perform validation and early stopping on a set of held-out, out-of-distribution synthetic functions. We leave this to future work.

### 3.2.1 Few-shot optimization for multiple objectives

As we mention in Section 2, one advantage of unsupervised pretraining is the ability to optimize for multiple objectives during the adaptation phase. In this section, we show that the same pretrained ExPT model is capable of optimizing different objectives in D'Kitty and Ant domains. We create two variants of the original D'Kitty, namely D'Kitty-45 and D'Kitty-60, whose objectives are to navigate the robot to goals that are $45°$ and $60°$ away from the original goal, respectively. For Ant, we create Ant-$v_y$ where the goal is to run as fast as possible in the vertical $Y$ direction (as opposed to horizontal $X$ direction in Ant) direction, and Ant-Energy, where the goal is to preserve energy. We detail how to construct these tasks in Appendix C. We use the same pretrained models for all these tasks. During adaptation, the model conditions on the $\mathcal{D}_{\text{few-shot}}$ and $y^\star$ for each task for optimization.

We evaluate ExPT on these tasks in the `poorest` setting. Table 3 shows that ExPT performs well on all tasks, where the median and mean scores are better than the best value in $\mathcal{D}_{\text{few-shot}}$, and the max score is close to 1. For the Ant, Ant-$v_y$, and Ant-Energy tasks, we visualize the behavior of the optimal designs that are discovered at `https://imgur.com/a/zpgI4YL`. When subject to the same policy-network (optimizing for horizontal $X$ speed), the robots optimized for different objectives behave differently; the optimal Ant is capable of leaping forward to move quickly in $X$; Ant-$v_y$ is able to jump up to maximize speed in $Y$; Ant-Energy is capable of 'sitting down' to conserve energy.

## 4 Related work

**Online ED** The majority of existing approaches solve ED in an active setting, where the model is allowed to query the black-box function to collect more data. Many of these works are based on Bayesian Optimization [39, 46, 50, 53, 56], which typically employs a surrogate model to the black-box function and an acquisition function. The surrogate model is often a predictive model that can quantify uncertainty, such as Gaussian Processes [54], Neural Processes [20, 21, 26, 29, 51, 45], or Bayesian Neural Networks [24]. The acquisition function uses the uncertainty output by the surrogate model to trade off between exploration and exploitation for querying new points.

**Offline ED**  Recent works have proposed to solve ED by learning from a fixed set of $(x, y)$ pairs to bypass active data collection [58, 36, 57, 12, 34, 16, 7, 15, 65]. The Design-Bench benchmark [58] consists of several such tasks in the physical sciences and robotics and is used by many recent works in offline ED. MINs [36] and BONET [34] perform optimization by generating designs $x$ via conditioning on a high score value $y$. MINs uses a GANs [25] model on $(x, y)$ pairs and BONET casts offline ED as a sequence modeling problem. COMs [57] formulates a conservative objective function that penalizes high-scoring poor designs and uses it to train a surrogate forward model which is then optimized using gradient ascent. BDI [12] uses a bidirectional model consisting of a forward and backward models that learn mappings from the dataset to high-scoring designs and vice versa. In contrast to these works, we propose ExPT in the few-shot ED setting, where the model is given access to only the $x's$ during pretraining, and a handful of labeled examples for adaptation.

**Synthetic Pretraining**  In the absence of vast amounts of labeled data, pretraining on synthetic data is an effective method for achieving significant gains in model performance. Prior works in this direction construct synthetic tasks which improve performance on diverse downstream tasks such as mathematical reasoning [64], text summarization [33], and perception tasks in vision [43]. Each synthetic task produces a dataset of labeled $(x, y)$ values that can be used to train a model as usual for various objectives. Often, pre-training in this manner produces better results than simply pre-training on another real dataset. In this work, we demonstrate that pretraining on synthetic data generated from GPs can achieve significant generalization to downstream functions, leading to state-of-the-art performance on challenging few-shot optimization problems.

**Few-shot learning**  Few-shot learning is a common paradigm in deep learning, where the model is pretrained on large amounts of data in an unsupervised manner. At test time, the model is given only a few examples from a downstream task and is expected to generalize [60]. This technique has found applications in text-generation (GPT-x) [9], image classification [55, 52], graph neural networks [18], text to visual-data generation [63], and neural architecture search [62] [10]. ExPT is capable of performing few-shot learning for black-box optimization in a variety of domains. Moreover, ExPT is pretrained on synthetically generated data with no prior knowledge of the downstream objective.

## 5   Conclusion

Inspired by real-world scenarios, this work introduces and studies the few-shot experimental design setting, where we aim to optimize a black-fox function given only a few examples. This setting is ubiquitous in many real-world applications, where experimental data collection is very expensive but we have access to unlabelled designs. We then propose ExPT, a foundation model style framework for few-shot experimental design. ExPT operates in two phases involving pretraining and finetuning. ExPT is pretrained on a rich family of synthetic functions using unlabeled data and can adapt to downstream functions with only a handful of data points via in-context learning. Empirically, ExPT outperforms all the existing methods by a large margin on all considered settings, especially improving over the second-best baseline by $70\%$ in the more challenging setting.

**Limitations and Future work**  In this work, we assume we have access to a larger unlabeled dataset for pretraining and the knowledge of the optimal value for optimization. While these assumptions are true in many applications and have been used widely in previous works, we would like to relax these assumptions in future work to improve further the applicability of the model. One more potential direction is to finetune the pretrained ExPT model on downstream data to further improve performance. Finally, we currently pretrain ExPT for each domain separately. We are interested in exploring if pretraining a big model that works for all domains is possible and if that helps improve performance in each individual domain.

## Acknowledgements

This work is supported by grants from Cisco, Meta, and Microsoft.

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

# A Additional experimental details

## A.1 Synthetic experiments

We pretrain the model for 2000 iterations with 128 synthetic functions at each iteration. We randomize the length scale parameter $\ell \sim \mathcal{U}[5.0, 10.0]$ and function scale parameter $\sigma_y \sim \mathcal{U}[1.0, 10.0]$ of the RBF kernel to increase pretraining data diversity. For each function generated, we sample 228 data points that we separate into 100 context points and 128 target points and train the model using the loss function in (2). Each input $x$ is a 32-dimensional vector, and each dimension is sampled from a uniform distribution $\mathcal{U}[-3, 3]$.

For each test function, we sample a large dataset of 20000 data points. We then randomly select 100 samples from the data points with function values lower than the 20th percentile as the few-shot data. We condition the model on this few-shot dataset and the maximal value $y^\star$ in the large dataset to generate 256 candidates and report the best score achieved among these candidates. We normalize the score to $[0, 1]$ using the worst and the best value in the large dataset.

## A.2 ExPT pretraining details

**Architectural details**    In all experiments, we use the same ExPT architecture. Before feeding to the Transformer encoder, we embed the $(y, x)$ context pairs with a 1-layer MLP and embed the target $y's$ with another 1-layer MLP. The transformer encoder has 4 layers with a hidden dimension of 128, 4 attention heads, `GELU` activation, and a dropout rate of 0.1. For the VAE model, we use a standard isotropic Gaussian distribution as the prior. Both the VAE encoder and VAE decoder have 4 layers with a hidden dimension of 512, and the latent variable $z$ has a dimension of 32.

**Optimization details**    In Design-Bench experiments, we pretrain ExPT for 10,000 iterations with 128 synthetic functions in each iteration. We use AdamW optimizer [31, 40] with a learning rate of $5e-4$ and $(\beta_1, \beta_2) = (0.9, 0.99)$. We use a linear warmup schedule for 1000 steps, followed by a cosine-annealing schedule for 9000 steps.

## A.3 Construction of new D'Kitty and Ant tasks

This section details how we constructed new objectives from the original D'Kitty and Ant that we used to evaluate ExPT in Section 3.2.1. For each new objective, we apply the corresponding oracle to the inputs $x's$ in the original dataset to create the dataset for the objective.

**Ant tasks**    In Ant, the original goal is to design a morphology that allows the Ant robot to run as fast as possible in the $x$ (horizontal) direction. The objective function is the sum of rewards in 100 time steps, where the reward $R$ at each time step is defined as:

$$R = \text{Forward reward} + \text{Survival reward} - \text{Control cost} - \text{Contact cost}, \tag{5}$$

where Forward reward $= (x_t - x_{t-1})/dt$ is the velocity of the Ant in the $x$ direction.

In Ant-$v_y$, the reward at each time step is similar, except that Forward reward $= (y_t - y_{t-1})/dt$ is the velocity of the Ant in the $y$ (vertical) direction. In other words, we aim to design morphologies that allow the robot to run fast in the $y$ direction.

In Ant-Energy, the reward at each time step is:

$$R = 1 + \text{Survival reward} - \text{Control cost} - \text{Contact cost}, \tag{6}$$

which means we incentivize the robot to conserve energy instead of running fast.

**D'Kitty tasks**    In D'Kitty, the goal is to design a morphology that allows the D'Kitty robot to reach a fixed target location, and the objective function $f$ is the Euclidean distance to the target. In the original D'Kitty task, the target location is on the vertical line from the starting point. In the two new tasks D'Kitty-45 and D'Kitty-60, the target locations are $45 \deg$ and $60 \deg$ away from the original target, respectively.

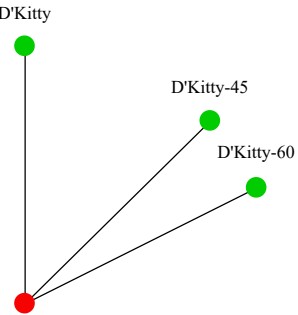

Figure 5: Different D'Kitty tasks. The red dot denotes the starting location, and the green dots are the target locations.

# B    Excluded Design-Bench tasks

## B.1    Superconductor

We found the approximate oracle provided by Design-Bench not accurate enough to provide a reliable comparison of optimization methods on this task. Figure 6 plots the score values in the dataset against the score values predicted by the approximate oracle, which shows a weak correlation between these two values.

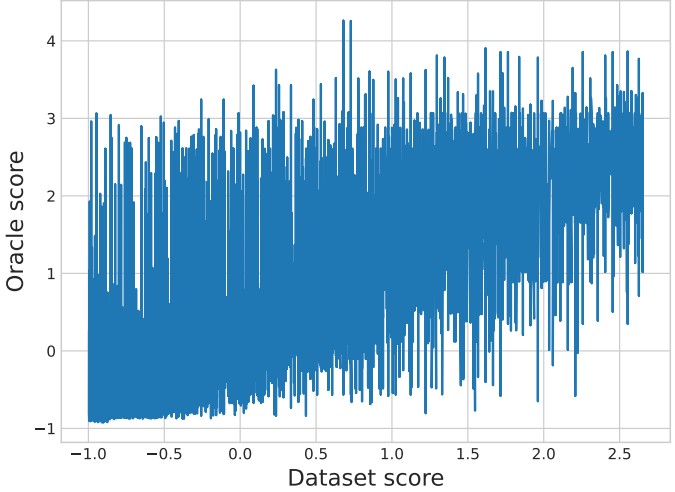

Figure 6: The correlation between the score values in the dataset ($x$-axis) and the score values predicted by the approximate oracle ($y$-axis) in Superconductor.

## B.2    Hopper

As noted in previous works that use Design-Bench [34], the oracle provided for the Hopper task is inconsistent with the true-dataset values. The outputs of the oracle on the dataset are skewed heavily towards low-function values, which makes it an unreliable task for evaluation.

## B.3    ChEMBL

As observed in previous works [58, 34], all methods produced nearly the same results on the ChEMBL task, so we excluded it in our experiments.

## C  Additional ablation and analysis

### C.1  Effects of GP hyperparameters

We empirically examine the impact of two GP hyperparameters, the variance $\sigma$ and the length scale $\ell$, on the performance of ExPT. Specifically, we evaluate the performance of ExPT on D'Kitty and Ant when $\sigma$ is too small (ExPT-small-$\sigma$) or too large (ExPT-large-$\sigma$), and when $\ell$ is too small (ExPT-small-$\ell$) or too large (ExPT-large-$\ell$). In ExPT-small-$\sigma$ and ExPT-large-$\sigma$, we sample $\sigma$ from $\mathcal{U}[0.01, 0.1]$ and $\mathcal{U}[100, 200]$, respectively. In ExPT-small-$\ell$ and ExPT-large-$\ell$, we sample $\ell$ from $\mathcal{U}[0.1, 1.0]$ and $\mathcal{U}[100, 200]$, respectively.

Table 4: Impact of $\sigma$ and $\ell$ on ExPT performance on Ant and D'Kitty in `random` (left) and `poorest` (right) settings. We average the performance across 3 seeds.

| | Baseline | D'Kitty | Ant | | Baseline | D'Kitty | Ant |
|---|---|---|---|---|---|---|---|
| | $\mathcal{D}_{\text{few-shot}}$(best) | 0.883 | 0.563 | | $\mathcal{D}_{\text{few-shot}}$(best) | 0.307 | 0.124 |
| Median | ExPT | $0.902 \pm 0.006$ | $\mathbf{0.705 \pm 0.018}$ | Median | ExPT | $\mathbf{0.922 \pm 0.009}$ | $\mathbf{0.686 \pm 0.090}$ |
| | ExPT-small-$\sigma$ | $\mathbf{0.915 \pm 0.006}$ | $0.661 \pm 0.111$ | | ExPT-small-$\sigma$ | $0.862 \pm 0.064$ | $0.656 \pm 0.098$ |
| | ExPT-large-$\sigma$ | $0.797 \pm 0.000$ | $0.471 \pm 0.012$ | | ExPT-large-$\sigma$ | $0.792 \pm 0.004$ | $0.489 \pm 0.019$ |
| | ExPT-small-$\ell$ | $0.793 \pm 0.004$ | $0.459 \pm 0.005$ | | ExPT-small-$\ell$ | $0.792 \pm 0.006$ | $0.462 \pm 0.004$ |
| | ExPT-large-$\ell$ | $0.795 \pm 0.003$ | $0.460 \pm 0.003$ | | ExPT-large-$\ell$ | $0.795 \pm 0.003$ | $0.460 \pm 0.004$ |
| Mean | ExPT | $0.865 \pm 0.016$ | $\mathbf{0.639 \pm 0.026}$ | Mean | ExPT | $\mathbf{0.871 \pm 0.018}$ | $\mathbf{0.646 \pm 0.061}$ |
| | ExPT-small-$\sigma$ | $\mathbf{0.896 \pm 0.016}$ | $0.630 \pm 0.089$ | | ExPT-small-$\sigma$ | $0.755 \pm 0.085$ | $0.606 \pm 0.077$ |
| | ExPT-large-$\sigma$ | $0.752 \pm 0.013$ | $0.534 \pm 0.015$ | | ExPT-large-$\sigma$ | $0.726 \pm 0.016$ | $0.547 \pm 0.012$ |
| | ExPT-small-$\ell$ | $0.726 \pm 0.018$ | $0.518 \pm 0.018$ | | ExPT-small-$\ell$ | $0.725 \pm 0.019$ | $0.529 \pm 0.014$ |
| | ExPT-large-$\ell$ | $0.725 \pm 0.016$ | $0.528 \pm 0.006$ | | ExPT-large-$\ell$ | $0.722 \pm 0.014$ | $0.530 \pm 0.011$ |

The results in Table 4 show that overall, suboptimal values of $\sigma$ and $\ell$ lead to a substantial drop in the performance of ExPT on both tasks. It is also noticeable that $\ell$ has a more significant influence on the performance than $\sigma$. In other words, the shape of the synthetic functions has a more critical impact on downstream performances than the magnitudes of the function values. A too small $\ell$ or large $\ell$ results in synthetic functions that exhibit either excessive oscillations or excessive smoothness, leading to poor generalization to downstream functions.

### C.2  ExPT with different pretraining data distributions

We perform an ablation study where we pretrain ExPT on different data distributions, including different GP kernels (GP-Cosine, GP-Linear, GP-Periodic), randomly initialized 1-layer neural networks (Random MLP), and neural network checkpoints trained on the few-shot data (Trained MLP). For each network used to generate data in Random MLP and Trained MLP, we randomly select the initialization method in {uniform, normal, xavier uniform, xavier normal, kaiming uniform, kaiming normal}, the hidden size in {16, 32, 64, 128, 256, 512, 1024}, and the depth in {2, 3, 4, 5, 6}. Each network in Random MLP is randomly initialized, while each network in Trained MLP is trained on the few-shot data.

Table 5: Performance of ExPT with different pretraining data distributions on the random setting

| | Pretraining data | D'Kitty | Ant | TF8 | TF10 | Mean score |
|---|---|---|---|---|---|---|
| Median | GP-RBF | $0.902 \pm 0.006$ | $0.705 \pm 0.018$ | $0.473 \pm 0.014$ | $0.477 \pm 0.014$ | $0.639 \pm 0.013$ |
| | GP-Cosine | $0.795 \pm 0.006$ | $0.463 \pm 0.003$ | $0.379 \pm 0.013$ | $0.456 \pm 0.006$ | $0.523 \pm 0.007$ |
| | GP-Linear | $0.900 \pm 0.002$ | $0.686 \pm 0.013$ | $0.377 \pm 0.009$ | $0.468 \pm 0.010$ | $0.608 \pm 0.009$ |
| | GP-Periodic | $0.902 \pm 0.003$ | $0.655 \pm 0.029$ | $0.452 \pm 0.013$ | $0.467 \pm 0.006$ | $0.619 \pm 0.013$ |
| | Random MLP | $0.906 \pm 0.004$ | $0.520 \pm 0.123$ | $0.480 \pm 0.021$ | $0.487 \pm 0.015$ | $0.598 \pm 0.041$ |
| | Trained MLP | $0.914 \pm 0.007$ | $0.691 \pm 0.003$ | $0.446 \pm 0.021$ | $0.482 \pm 0.029$ | $0.633 \pm 0.015$ |
| Max | GP-RBF | $0.973 \pm 0.005$ | $0.970 \pm 0.004$ | $0.933 \pm 0.036$ | $0.677 \pm 0.048$ | $0.888 \pm 0.023$ |
| | GP-Cosine | $0.955 \pm 0.008$ | $0.963 \pm 0.011$ | $0.906 \pm 0.079$ | $0.709 \pm 0.068$ | $0.883 \pm 0.042$ |
| | GP-Linear | $0.972 \pm 0.001$ | $0.965 \pm 0.016$ | $0.899 \pm 0.095$ | $0.654 \pm 0.033$ | $0.872 \pm 0.036$ |
| | GP-Periodic | $0.971 \pm 0.005$ | $0.966 \pm 0.005$ | $0.875 \pm 0.022$ | $0.646 \pm 0.026$ | $0.864 \pm 0.014$ |
| | Random MLP | $0.973 \pm 0.001$ | $0.953 \pm 0.013$ | $0.938 \pm 0.068$ | $0.653 \pm 0.004$ | $0.879 \pm 0.022$ |
| | Trained MLP | $0.974 \pm 0.005$ | $0.935 \pm 0.022$ | $0.879 \pm 0.039$ | $0.660 \pm 0.003$ | $0.862 \pm 0.017$ |
| Mean | GP-RBF | $0.865 \pm 0.016$ | $0.639 \pm 0.026$ | $0.476 \pm 0.010$ | $0.474 \pm 0.015$ | $0.614 \pm 0.017$ |
| | GP-Cosine | $0.725 \pm 0.022$ | $0.534 \pm 0.011$ | $0.385 \pm 0.007$ | $0.455 \pm 0.004$ | $0.525 \pm 0.011$ |
| | GP-Linear | $0.866 \pm 0.001$ | $0.633 \pm 0.017$ | $0.397 \pm 0.013$ | $0.465 \pm 0.010$ | $0.590 \pm 0.010$ |
| | GP-Periodic | $0.865 \pm 0.008$ | $0.594 \pm 0.010$ | $0.464 \pm 0.008$ | $0.469 \pm 0.008$ | $0.598 \pm 0.009$ |
| | Random MLP | $0.883 \pm 0.011$ | $0.516 \pm 0.074$ | $0.481 \pm 0.016$ | $0.485 \pm 0.016$ | $0.591 \pm 0.029$ |
| | Trained MLP | $0.910 \pm 0.008$ | $0.660 \pm 0.003$ | $0.451 \pm 0.019$ | $0.478 \pm 0.026$ | $0.625 \pm 0.014$ |

Table 6: Performance of ExPT with different pretraining data distributions on the poor setting

| | Pretraining data | D'Kitty | Ant | TF8 | TF10 | Mean score |
|---|---|---|---|---|---|---|
| | GP-RBF | 0.922 ± 0.009 | 0.686 ± 0.090 | 0.552 ± 0.042 | 0.489 ± 0.013 | 0.662 ± 0.039 |
| | GP-Cosine | 0.795 ± 0.005 | 0.463 ± 0.003 | 0.379 ± 0.013 | 0.456 ± 0.006 | 0.524 ± 0.007 |
| Median | GP-Linear | 0.918 ± 0.009 | 0.675 ± 0.065 | 0.380 ± 0.013 | 0.450 ± 0.004 | 0.606 ± 0.023 |
| | GP-Periodic | 0.928 ± 0.006 | 0.689 ± 0.037 | 0.487 ± 0.089 | 0.498 ± 0.013 | 0.651 ± 0.036 |
| | Random MLP | 0.902 ± 0.012 | 0.446 ± 0.004 | 0.499 ± 0.010 | 0.495 ± 0.005 | 0.586 ± 0.008 |
| | Trained MLP | 0.909 ± 0.006 | 0.733 ± 0.039 | 0.431 ± 0.043 | 0.482 ± 0.028 | 0.639 ± 0.029 |
| | GP-RBF | 0.946 ± 0.018 | 0.965 ± 0.004 | 0.873 ± 0.035 | 0.615 ± 0.022 | 0.850 ± 0.020 |
| | GP-Cosine | 0.961 ± 0.004 | 0.951 ± 0.027 | 0.906 ± 0.079 | 0.709 ± 0.068 | 0.872 ± 0.045 |
| Max | GP-Linear | 0.976 ± 0.003 | 0.971 ± 0.008 | 0.896 ± 0.012 | 0.623 ± 0.030 | 0.867 ± 0.013 |
| | GP-Periodic | 0.975 ± 0.004 | 0.969 ± 0.001 | 0.709 ± 0.086 | 0.641 ± 0.061 | 0.824 ± 0.038 |
| | Random MLP | 0.975 ± 0.003 | 0.970 ± 0.007 | 0.797 ± 0.050 | 0.629 ± 0.018 | 0.843 ± 0.020 |
| | Trained MLP | 0.975 ± 0.003 | 0.905 ± 0.033 | 0.716 ± 0.094 | 0.578 ± 0.023 | 0.794 ± 0.038 |
| | GP-RBF | 0.871 ± 0.018 | 0.646 ± 0.061 | 0.549 ± 0.032 | 0.488 ± 0.011 | 0.639 ± 0.031 |
| | GP-Cosine | 0.728 ± 0.021 | 0.528 ± 0.010 | 0.385 ± 0.007 | 0.455 ± 0.004 | 0.524 ± 0.010 |
| Mean | GP-Linear | 0.872 ± 0.025 | 0.624 ± 0.031 | 0.397 ± 0.009 | 0.447 ± 0.004 | 0.585 ± 0.017 |
| | GP-Periodic | 0.887 ± 0.047 | 0.634 ± 0.015 | 0.511 ± 0.069 | 0.496 ± 0.011 | 0.634 ± 0.036 |
| | Random MLP | 0.790 ± 0.048 | 0.522 ± 0.042 | 0.499 ± 0.012 | 0.489 ± 0.006 | 0.575 ± 0.027 |
| | Trained MLP | 0.869 ± 0.012 | 0.684 ± 0.043 | 0.447 ± 0.057 | 0.476 ± 0.027 | 0.619 ± 0.022 |

Tables 5 and 6 show the performance of ExPT on the few-shot random and few-shot poor settings when pretrained with different data distributions. Overall, the model achieves good performance across different data distributions, with GP-RBF being the best in most settings. This ablation study shows the robustness of ExPT to the pretraining data distribution.

## C.3 ExPT with different decoder architectures

In addition to the pretraining data distribution, we also conducted an ablation study on the architecture of ExPT, in which we replaced the VAE model with a diffusion model (ExPT-Diffusion). We take the diffusion architecture from [35].

Table 7: Performance of ExPT with different decoder architectures on the random setting

| | Decoder architecture | D'Kitty | Ant | TF8 | TF10 | Mean score |
|---|---|---|---|---|---|---|
| Median | VAE | 0.902 ± 0.006 | 0.705 ± 0.018 | 0.473 ± 0.014 | 0.477 ± 0.014 | 0.639 ± 0.013 |
| | Diffusion | 0.816 ± 0.028 | 0.642 ± 0.018 | 0.457 ± 0.116 | 0.489 ± 0.019 | 0.601 ± 0.045 |
| Max | VAE | 0.973 ± 0.005 | 0.970 ± 0.004 | 0.933 ± 0.036 | 0.677 ± 0.048 | 0.888 ± 0.023 |
| | Diffusion | 0.966 ± 0.007 | 0.967 ± 0.006 | 0.868 ± 0.150 | 0.628 ± 0.014 | 0.857 ± 0.044 |
| Mean | VAE | 0.865 ± 0.016 | 0.639 ± 0.026 | 0.476 ± 0.010 | 0.474 ± 0.015 | 0.614 ± 0.017 |
| | Diffusion | 0.741 ± 0.013 | 0.603 ± 0.016 | 0.468 ± 0.115 | 0.486 ± 0.016 | 0.575 ± 0.040 |

Table 8: Performance of ExPT with different decoder architectures on the poor setting

| | Decoder architecture | D'Kitty | Ant | TF8 | TF10 | Mean score |
|---|---|---|---|---|---|---|
| Median | VAE | 0.922 ± 0.009 | 0.686 ± 0.090 | 0.552 ± 0.042 | 0.489 ± 0.013 | 0.662 ± 0.039 |
| | Diffusion | 0.821 ± 0.038 | 0.638 ± 0.011 | 0.295 ± 0.010 | 0.421 ± 0.007 | 0.544 ± 0.017 |
| Max | VAE | 0.946 ± 0.018 | 0.965 ± 0.004 | 0.873 ± 0.035 | 0.615 ± 0.022 | 0.850 ± 0.020 |
| | Diffusion | 0.974 ± 0.003 | 0.956 ± 0.008 | 0.677 ± 0.007 | 0.593 ± 0.026 | 0.800 ± 0.011 |
| Mean | VAE | 0.871 ± 0.018 | 0.646 ± 0.061 | 0.549 ± 0.032 | 0.488 ± 0.011 | 0.639 ± 0.031 |
| | Diffusion | 0.731 ± 0.035 | 0.600 ± 0.014 | 0.311 ± 0.011 | 0.415 ± 0.013 | 0.514 ± 0.018 |

Tables 7 and 8 show that ExPT + VAE outperforms ExPT + Diffusion in all tasks and settings. We hypothesize that ExPT with a too powerful decoder may learn only to model the distribution over the target $x's$ and ignore the conditioning variables (context $x's$, context $y's$, and target $y$), which consequently hurts the generalization of the model.

### C.3.1 Forward modeling versus Inverse modeling

As we mentioned in Section 2.2, two possible approaches exist to pretrain ExPT on synthetic data. We take the inverse modeling approach for ExPT throughout the paper, as we train ExPT to directly produce design inputs $x's$. In this section, we empirically validate our design choices by comparing ExPT with TNP-ED, its forward counterpart. TNP-ED's architecture is similar to ExPT's in Figure 2, except that the target points now contain $x_{m+1:N}$ instead of $y_{m+1:N}$, the decoder is a 1-layer MLP,

and the predicted outputs are $\hat{y}_{m+1:N}$. We call this model TNP-ED because a model named TNP [45] with a similar architecture was previously proposed in the context of meta-learning. We pretrain TNP-ED using a simple mean-squared error loss $\mathcal{L} = \sum_{i=m+1}^{N}(\hat{y}_i - y_i)^2$. After pretraining, we condition TNP-ED on $\mathcal{D}_{\text{few-shot}}$ and the best inputs in this dataset, and perform gradient ascent with respect to these inputs to obtain better points.

Table 9: Comparison of inverse modeling (ExPT) versus forward modeling (TNP-ED) on Ant and D'Kitty in `random` (left) and `poorest` (right) settings. We average the performance across 3 seeds.

| | Baseline | D'Kitty | Ant | | Baseline | D'Kitty | Ant |
|---|---|---|---|---|---|---|---|
| | $\mathcal{D}_{\text{few-shot}}$(best) | 0.883 | 0.563 | | $\mathcal{D}_{\text{few-shot}}$(best) | 0.307 | 0.124 |
| Median | ExPT | **0.902 ± 0.006** | **0.705 ± 0.018** | Median | ExPT | **0.922 ± 0.009** | **0.686 ± 0.090** |
| | TNP-ED | 0.770 ± 0.009 | 0.438 ± 0.007 | | TNP-ED | 0.309 ± 0.000 | 0.197 ± 0.005 |
| Mean | ExPT | **0.865 ± 0.016** | **0.639 ± 0.026** | Mean | ExPT | **0.871 ± 0.018** | **0.646 ± 0.061** |
| | TNP-ED | 0.670 ± 0.037 | 0.451 ± 0.018 | | TNP-ED | 0.405 ± 0.004 | 0.237 ± 0.005 |

Table 9 compares the performance of ExPT and TNP-ED on D'Kitty and Ant with respect to the median score and mean score. ExPT achieves significantly better performance in all metrics, especially in the `poorest` setting. This is because forward models suffer from poor out-of-distribution generalization, and performing gradient ascent on this model may result in points that have high values under the model but are very poor when evaluated using the true functions. This validates our inverse modeling approach.

## C.4 ExPT with sequential sampling

A significant advantage of ExPT is its ability to adapt to any objective function purely through in-context learning. This means that the model can refine its understanding of the underlying objective function given more data points in a very efficient manner. In this section, we explore an alternative optimization scheme for ExPT, namely *sequential sampling*, which explicitly utilizes the in-context learning ability of the model. Specifically, instead of producing $Q = 256$ inputs simultaneously, we sample one by one sequentially. That is, we condition the model on $\mathcal{D}_{\text{few-shot}}$ and $y^\star$ to sample the first point, evaluate the point using the black-box function, and add the point together with its score to the context set. We repeat this process for 256 times.

Table 10: Comparison of simultaneous (ExPT) and sequential (ExPT-Seq) sampling on Ant and D'Kitty in `random` (left) and `poorest` (right) settings. We average the performance across 3 seeds.

| | Baseline | D'Kitty | Ant | | Baseline | D'Kitty | Ant |
|---|---|---|---|---|---|---|---|
| | $\mathcal{D}_{\text{few-shot}}$(best) | 0.883 | 0.563 | | $\mathcal{D}_{\text{few-shot}}$(best) | 0.307 | 0.124 |
| Median | ExPT | 0.902 ± 0.006 | 0.705 ± 0.018 | Median | ExPT | 0.922 ± 0.009 | 0.686 ± 0.090 |
| | ExPT-Seq | **0.903 ± 0.005** | **0.719 ± 0.013** | | ExPT-Seq | **0.928 ± 0.012** | **0.822 ± 0.055** |
| Mean | ExPT | 0.865 ± 0.016 | 0.639 ± 0.026 | Mean | ExPT | 0.871 ± 0.018 | 0.646 ± 0.061 |
| | ExPT-Seq | **0.872 ± 0.010** | **0.669 ± 0.017** | | ExPT-Seq | **0.923 ± 0.011** | **0.767 ± 0.048** |

Table 10 shows that ExPT with sequential sampling performs better than simultaneous sampling on D'Kitty and Ant in both `random` and `poor` settings. Especially on Ant in the `poorest` setting, ExPT-Sequential achieves improvements of 20% and 19% over ExPT in terms of the median and mean performance, respectively. Intuitively, as we add more data points to the context set, ExPT-Sequential is able to updates its understanding of the structure of the objective function, consequently leading to improved performance.

## C.5 Effects of $|\mathcal{D}_{\text{unlabeled}}|$

We empirically examine the effects of the size of $\mathcal{D}_{\text{unlabeled}}$ on the downstream performance of ExPT. Specifically, we subsample the $x's$ in the `public` dataset with a ratio $r \in \{0.01, 0.1, 0.2, 0.5, 1.0\}$. Adaptation and evaluation are the same as in Section 3.

Figure 7 shows the median and mean performance of ExPT on Dkitty and Ant in both `random` and `poorest` settings with respect to the ratio $r$. In the `random` setting, ExPT is able to reach or surpass the best data point in the few-shot dataset by using as few as 0.2 of the pretraining data. In the `poorest` setting, ExPT performs better than the best dataset point with only 0.01 of the

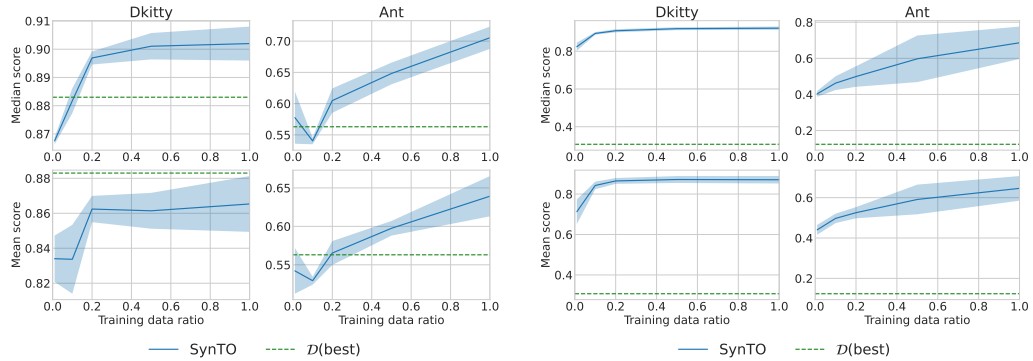

Figure 7: The performance of ExPT on Dkitty and Ant in the `random` (left) and `poorest` (right) setting when we vary the training data ration $r$. We average the performance across 3 seeds.

pretraining data. Moreover, the performance improves consistently as the pretraining data size increases, suggesting that we can achieve even better performance by simply using more unlabeled data for pretraining. This result highlights the unique capability of ExPT of learning from unlabeled data, providing new opportunities for solving challenging optimization problems where unlabeled data is plentiful but labeled data is scarce.

## C.6 Sorting context and target points

In the main experiments in Section 3, for each generated function during pretrainnig, we sample 228 points that we divide randomly into 100 context points and 128 target points. However, at adaptation, we condition on target output values that are likely to be higher than the best input value in the context set. Therefore, it is natural to sort the context points and target points during pretraining, so that the target inputs always have higher values than the context inputs. We denote this pretraining mechanism as ExPT-sorted.

Table 11: Comparison of pretraining on randomly divided context and target points (ExPT) versus sorted context and target points (ExPT-sorted) on Ant and D'Kitty in `random` (left) and `poorest` (right) settings. We average the performance across 3 seeds.

| | Baseline | D'Kitty | Ant | | Baseline | D'Kitty | Ant |
|---|---|---|---|---|---|---|---|
| | $\mathcal{D}_{\text{few-shot}}(\text{best})$ | 0.883 | 0.563 | | $\mathcal{D}_{\text{few-shot}}(\text{best})$ | 0.307 | 0.124 |
| Median | ExPT | $\mathbf{0.902 \pm 0.006}$ | $\mathbf{0.705 \pm 0.018}$ | Median | ExPT | $\mathbf{0.922 \pm 0.009}$ | $\mathbf{0.686 \pm 0.090}$ |
| | ExPT-Sorted | $0.811 \pm 0.019$ | $0.631 \pm 0.015$ | | ExPT-Sorted | $0.911 \pm 0.003$ | $0.685 \pm 0.044$ |
| Mean | ExPT | $\mathbf{0.865 \pm 0.016}$ | $\mathbf{0.639 \pm 0.026}$ | Mean | ExPT | $0.871 \pm 0.018$ | $\mathbf{0.646 \pm 0.061}$ |
| | ExPT-Sorted | $0.794 \pm 0.020$ | $0.590 \pm 0.014$ | | ExPT-Sorted | $\mathbf{0.900 \pm 0.003}$ | $0.642 \pm 0.035$ |

Table 11 shows that ExPT-sorted underperforms ExPT in the `random` setting, while performing very similarly in the `poorest` setting. This indicates that learning to predict any points provides a better and more general pretraining objective than only learning to predict points with high values.

## C.7 Comparisons with more baselines

In addition to the baselines in Section 3, we compare ExPT with 3 variants of Gradient Ascent, a method that was considered in previous works [58, 36, 57, 34]. The Grad. Asc baseline simply learns a forward model and finds an optimal $x^*$ by taking 200 gradient-ascent steps to improve an existing input $x$. The two variants Grad. Min and Grad. Mean create ensembles of forward models and perform gradient ascent using the min and mean ensemble predictions.

Tables 12 and 13 show the performance of ExPT and all baselines in the `random` and `poorest` settings. We see that while the gradient ascent methods perform well on certain tasks, with good performance on the TF-Bind8 task in particular, ExPT is still the best performing method in all settings and metrics.

Table 12: Comparison of ExPT and the baselines on the few-shot `random` setting of 4 Design-Bench tasks. We report median, max, and mean performance across 3 random seeds. Higher scores are better. Blue denotes the best entry in the column, and Violet denotes the second best.

| | Baseline | D'Kitty | Ant | TF Bind 8 | TF Bind 10 | Mean score (↑) |
|---|---|---|---|---|---|---|
| | $\mathcal{D}_{\text{few-shot}}$(best) | 0.883 | 0.563 | 0.439 | 0.466 | — |
| Median | MINs | $0.859 \pm 0.014$ | $0.485 \pm 0.152$ | $0.416 \pm 0.019$ | $0.468 \pm 0.014$ | $0.557 \pm 0.050$ |
| | COMs | $0.752 \pm 0.007$ | $0.411 \pm 0.012$ | $0.371 \pm 0.001$ | $0.468 \pm 0.000$ | $0.501 \pm 0.005$ |
| | BONET | $0.852 \pm 0.013$ | $0.597 \pm 0.119$ | $0.441 \pm 0.003$ | $0.483 \pm 0.009$ | $0.593 \pm 0.036$ |
| | BDI | $0.592 \pm 0.020$ | $0.396 \pm 0.018$ | $0.540 \pm 0.032$ | $0.438 \pm 0.034$ | $0.492 \pm 0.026$ |
| | GP-qEI | $0.842 \pm 0.058$ | $0.550 \pm 0.007$ | $0.439 \pm 0.000$ | $0.467 \pm 0.000$ | $0.575 \pm 0.016$ |
| | Grad. Asc | $0.403 \pm 0.134$ | $0.088 \pm 0.017$ | $0.492 \pm 0.017$ | $0.492 \pm 0.018$ | $0.369 \pm 0.0465$ |
| | Grad. Min | $0.712 \pm 0.028$ | $0.220 \pm 0.035$ | $0.504 \pm 0.025$ | $0.465 \pm 0.008$ | $0.475 \pm 0.024$ |
| | Grad. Mean | $0.437 \pm 0.180$ | $0.150 \pm 0.037$ | $0.551 \pm 0.029$ | $0.485 \pm 0.018$ | $0.406 \pm 0.066$ |
| | ExPT | $0.902 \pm 0.006$ | $0.705 \pm 0.018$ | $0.473 \pm 0.014$ | $0.477 \pm 0.014$ | $0.639 \pm 0.013$ |
| Max | MINs | $0.930 \pm 0.010$ | $0.890 \pm 0.017$ | $0.814 \pm 0.030$ | $0.639 \pm 0.017$ | $0.818 \pm 0.019$ |
| | COMs | $0.920 \pm 0.010$ | $0.841 \pm 0.044$ | $0.686 \pm 0.152$ | $0.656 \pm 0.020$ | $0.776 \pm 0.057$ |
| | BONET | $0.909 \pm 0.012$ | $0.888 \pm 0.024$ | $0.887 \pm 0.053$ | $0.702 \pm 0.006$ | $0.847 \pm 0.024$ |
| | BDI | $0.918 \pm 0.006$ | $0.806 \pm 0.094$ | $0.906 \pm 0.074$ | $0.532 \pm 0.023$ | $0.791 \pm 0.049$ |
| | GP-qEI | $0.896 \pm 0.000$ | $0.887 \pm 0.000$ | $0.513 \pm 0.104$ | $0.647 \pm 0.011$ | $0.736 \pm 0.029$ |
| | Grad. Asc | $0.775 \pm 0.032$ | $0.240 \pm 0.032$ | $0.923 \pm 0.005$ | $0.675 \pm 0.017$ | $0.653 \pm 0.0215$ |
| | Grad. Min | $0.822 \pm 0.053$ | $0.434 \pm 0.092$ | $0.960 \pm 0.002$ | $0.632 \pm 0.009$ | $0.712 \pm 0.039$ |
| | Grad. Mean | $0.829 \pm 0.009$ | $0.337 \pm 0.063$ | $0.957 \pm 0.010$ | $0.668 \pm 0.034$ | $0.698 \pm 0.029$ |
| | ExPT | $0.973 \pm 0.005$ | $0.970 \pm 0.004$ | $0.933 \pm 0.036$ | $0.677 \pm 0.048$ | $0.888 \pm 0.023$ |
| Mean | MINs | $0.624 \pm 0.025$ | $0.009 \pm 0.013$ | $0.415 \pm 0.030$ | $0.465 \pm 0.015$ | $0.378 \pm 0.021$ |
| | COMs | $0.515 \pm 0.050$ | $0.020 \pm 0.006$ | $0.369 \pm 0.003$ | $0.471 \pm 0.004$ | $0.344 \pm 0.016$ |
| | BONET | $0.837 \pm 0.023$ | $0.579 \pm 0.024$ | $0.448 \pm 0.011$ | $0.484 \pm 0.009$ | $0.587 \pm 0.017$ |
| | BDI | $0.570 \pm 0.032$ | $0.385 \pm 0.012$ | $0.536 \pm 0.032$ | $0.444 \pm 0.027$ | $0.484 \pm 0.026$ |
| | GP-qEI | $0.505 \pm 0.006$ | $0.019 \pm 0.001$ | $0.439 \pm 0.001$ | $0.473 \pm 0.002$ | $0.359 \pm 0.003$ |
| | Grad. Asc | $0.400 \pm 0.073$ | $0.090 \pm 0.018$ | $0.513 \pm 0.014$ | $0.492 \pm 0.017$ | $0.374 \pm 0.031$ |
| | Grad. Min | $0.599 \pm 0.068$ | $0.221 \pm 0.034$ | $0.531 \pm 0.015$ | $0.462 \pm 0.009$ | $0.453 \pm 0.032$ |
| | Grad. Mean | $0.527 \pm 0.079$ | $0.150 \pm 0.038$ | $0.569 \pm 0.028$ | $0.438 \pm 0.017$ | $0.421 \pm 0.041$ |
| | ExPT | $0.865 \pm 0.016$ | $0.639 \pm 0.026$ | $0.476 \pm 0.010$ | $0.474 \pm 0.015$ | $0.614 \pm 0.017$ |

Table 13: Comparison of ExPT and the baselines on the few-shot `poorest` setting of 4 Design-Bench tasks. We report the median, max, and mean performance across 3 random seeds. Higher scores are better. Blue denotes the best entry in the column, and Violet denotes the second best.

| | Baseline | D'Kitty | Ant | TF Bind 8 | TF Bind 10 | Mean score (↑) |
|---|---|---|---|---|---|---|
| | $\mathcal{D}_{\text{few-shot}}$(best) | 0.307 | 0.124 | 0.124 | 0.000 | — |
| Median | MINs | $0.480 \pm 0.156$ | $0.316 \pm 0.040$ | $0.437 \pm 0.007$ | $0.463 \pm 0.003$ | $0.424 \pm 0.052$ |
| | COMs | $0.733 \pm 0.023$ | $0.401 \pm 0.026$ | $0.111 \pm 0.000$ | $0.459 \pm 0.006$ | $0.426 \pm 0.014$ |
| | BONET | $0.310 \pm 0.000$ | $0.236 \pm 0.047$ | $0.319 \pm 0.018$ | $0.461 \pm 0.017^{*}$ | $0.332 \pm 0.021$ |
| | BDI | $0.309 \pm 0.000$ | $0.192 \pm 0.012$ | $0.365 \pm 0.000$ | $0.454 \pm 0.017$ | $0.330 \pm 0.007$ |
| | GP-qEI | $0.883 \pm 0.000$ | $0.565 \pm 0.001$ | $0.439 \pm 0.000$ | $0.467 \pm 0.000$ | $0.589 \pm 0.000$ |
| | Grad. Asc | $0.741 \pm 0.026$ | $0.321 \pm 0.073$ | $0.425 \pm 0.064$ | $0.419 \pm 0.073$ | $0.477 \pm 0.044$ |
| | Grad. Min | $0.806 \pm 0.004$ | $0.454 \pm 0.061$ | $0.357 \pm 0.040$ | $0.376 \pm 0.079$ | $0.498 \pm 0.046$ |
| | Grad. Mean | $0.742 \pm 0.054$ | $0.472 \pm 0.066$ | $0.350 \pm 0.014$ | $0.395 \pm 0.019$ | $0.489 \pm 0.038$ |
| | ExPT | $0.922 \pm 0.009$ | $0.686 \pm 0.090$ | $0.552 \pm 0.042$ | $0.489 \pm 0.013$ | $0.662 \pm 0.039$ |
| Max | MINs | $0.841 \pm 0.014$ | $0.721 \pm 0.031$ | $0.962 \pm 0.019$ | $0.648 \pm 0.025$ | $0.793 \pm 0.022$ |
| | COMs | $0.931 \pm 0.022$ | $0.843 \pm 0.020$ | | $0.739 \pm 0.057$ | $0.659 \pm 0.025$ |
| | BONET | $0.929 \pm 0.031$ | $0.557 \pm 0.118$ | $0.809 \pm 0.038$ | $0.519 \pm 0.039^{*}$ | $0.704 \pm 0.057$ |
| | BDI | $0.939 \pm 0.002$ | $0.693 \pm 0.109$ | $0.913 \pm 0.000$ | $0.596 \pm 0.020$ | $0.785 \pm 0.033$ |
| | GP-qEI | $0.896 \pm 0.000$ | $0.887 \pm 0.000$ | $0.439 \pm 0.000$ | $0.645 \pm 0.021$ | $0.717 \pm 0.005$ |
| | Grad. Asc | $0.837 \pm 0.038$ | $0.684 \pm 0.071$ | $0.821 \pm 0.077$ | $0.568 \pm 0.019$ | $0.728 \pm 0.052$ |
| | Grad. Min | $0.910 \pm 0.009$ | $0.801 \pm 0.029$ | $0.842 \pm 0.066$ | $0.555 \pm 0.028$ | $0.777 \pm 0.033$ |
| | Grad. Mean | $0.882 \pm 0.028$ | $0.807 \pm 0.046$ | $0.747 \pm 0.055$ | $0.542 \pm 0.039$ | $0.745 \pm 0.042$ |
| | ExPT | $0.946 \pm 0.018$ | $0.965 \pm 0.004$ | $0.873 \pm 0.035$ | $0.615 \pm 0.022$ | $0.850 \pm 0.020$ |
| Mean | MINs | $0.623 \pm 0.051$ | $0.015 \pm 0.017$ | $0.464 \pm 0.009$ | $0.463 \pm 0.002$ | $0.391 \pm 0.020$ |
| | COMs | $0.607 \pm 0.021$ | $0.033 \pm 0.003$ | $0.109 \pm 0.001$ | $0.454 \pm 0.004$ | $0.301 \pm 0.007$ |
| | BONET | $0.490 \pm 0.023$ | $0.234 \pm 0.052$ | $0.318 \pm 0.018$ | $0.459 \pm 0.018$ | $0.375 \pm 0.028$ |
| | BDI | $0.364 \pm 0.004$ | $0.215 \pm 0.021$ | $0.369 \pm 0.000$ | $0.453 \pm 0.018$ | $0.350 \pm 0.011$ |
| | GP-qEI | $0.533 \pm 0.001$ | $0.018 \pm 0.000$ | $0.439 \pm 0.000$ | $0.470 \pm 0.002$ | $0.365 \pm 0.001$ |
| | Grad. Asc | $0.659 \pm 0.069$ | $0.334 \pm 0.018$ | $0.432 \pm 0.061$ | $0.427 \pm 0.042$ | $0.463 \pm 0.048$ |
| | Grad. Min | $0.794 \pm 0.003$ | $0.454 \pm 0.051$ | $0.374 \pm 0.018$ | $0.386 \pm 0.044$ | $0.502 \pm 0.029$ |
| | Grad. Mean | $0.702 \pm 0.083$ | $0.467 \pm 0.050$ | $0.356 \pm 0.023$ | $0.405 \pm 0.018$ | $0.483 \pm 0.044$ |
| | ExPT | $0.871 \pm 0.018$ | $0.646 \pm 0.061$ | $0.549 \pm 0.032$ | $0.488 \pm 0.011$ | $0.639 \pm 0.031$ |

# D  Compute

All training is done on 10 AMD EPYC 7313 CPU cores and one NVIDIA RTX A5000 GPU.

# E Reproducibility

We made a strong effort to ensure that our work can be reproduced properly. In Section 2, we provide a comprehensive description of our methodology, while in Section 3 and Appendix A, we provide the specifics of our pretraining and evaluation setup, as well as our choice of hyperparameters. We compare our approach with various baseline methods from different approaches on multiple tasks in Design-Bench [58] with distinct properties. Our results are averaged over 3 seeds and we also report the standard deviation. Additionally, we conduct several ablation experiments to examine how sensitive ExPT is to different hyperparameters.

# F Broader impact

The field of offline black-box optimization can have positive impacts in many spheres, including in drug-discovery, nuclear reactor design, and optimal robot design. The few-shot setting that we introduce in this work is also highly relevant to these fields which have large quantities of unlabelled data, but only a limited quantity of labelled data points. It is also worth noting however, that it is possible to use black-box optimization in general for malicious purposes such as to produce chemicals with harmful properties. Even though our work does not directly enable such use cases, this possibility should be taken into account when applying ExPT and similar frameworks to these kinds of impactful real-world scenarios.

