# OpenReview forum: "ExPT: Synthetic Pretraining for Few-Shot Experimental Design"
_NeurIPS.cc/2023/Workshop/AI4Science — NeurIPS2023-AI4Science Poster_

### Official Review · Reviewer_9Gjn · 2023-10-25
**Interesting approach for few-shot Experimental Design problem by synthetic pretraining with an unlabeled dataset and then adapt the model to the downstream task with only a handful of labeled examples.**

**Rating:** 6
**Confidence:** 3

**Review:**

Please find my comments below:

a) The proposed method is interesting and is shown to perform better in tasks, set in random and poorest settings, compared to previous approaches. Even though the four different data sets are considered to show the effectiveness of the proposed method, they only seem to be a pair of datasets as D'Kitty and Ant are very similar in terms of goals, and the same goes for TF Bind 8 and 10. The authors should have considered datasets from other domains, such as image, to prove the efficacy of their approach.

b) The decoder of the proposed method is a VAE that generates data, conditioned on the hidden vectors generated from the encoder transformer. But this terminologies are misleading as VAE itself is an encoder-decoder architecture. Hence it should be clearly mentioned in the explanation and in Fig1.

c) The proposed few-shot generalization and optimization through in-context learning is gradient-free, but were not clearly explained.

d) In L217, under pretraining step for 10000 iterations with 128 synthetic functions in each iterations, how did it result in 1280000 synthetic functions? Why one function would be considered different ones under multiple iterations?

---

### Meta-Review · Area_Chair_oCxz · 2023-10-27

**Recommendation:** Accept (Poster)
**Confidence:** 3

**Metareview:**

The author introduces the Pretrained Transformers for Experimental Design (ExPT). This model uniquely blends synthetic pretraining with ICL to facilitate few-shot learning. I urge the authors to carefully address the concerns raised by the reviewers. In particular, section 3.2 is challenging to understand, and Figure 1 does not adequately clarify, especially with regard to the role of the VAE. However, I believe that these concerns can be adequately addressed through a thorough revision.